# PeerJ

# Childhood socioeconomic deprivation, but not current mood, is associated with behavioural disinhibition in adults

Tünde Paál[1,2,3], Thomas Carpenter[1,3] and Daniel Nettle[1]

[1] Centre for Behaviour and Evolution & Institute of Neuroscience, Newcastle University, Newcastle, UK
[2] University of Pécs, Institute of Psychology, Pécs, Hungary
[3] These authors contributed equally to this work.

## ABSTRACT

There is evidence to suggest that impulsivity is predicted by socioeconomic background, with people from more deprived backgrounds tending to be more impulsive, and by current mood, with poorer mood associated with greater impulsivity. However, impulsivity is not a unitary construct, and previous research in this area has focused on measures of 'waiting' impulsivity rather than behavioural disinhibition. We administered a standard measure of behavioural disinhibition, the stop-signal task, to 58 adult participants from a community sample. We had measured socioeconomic background using participant postcode at age 16, and assigned participants to receive either a neutral or a negative mood induction. We found no effects of mood on behavioural disinhibition, but we found a significant effect of socioeconomic background. Participants who had lived in more deprived postcodes at age 16 showed longer stop-signal reaction times, and hence greater behavioural disinhibition. The pattern was independent of participant age and overall reaction time. Though caution is required inferring causality from correlation, it is possible that that experiencing socioeconomic deprivation in childhood and adolescence may lead to greater behavioural disinhibition in adulthood.

## INTRODUCTION

Impulsivity is an important psychological construct, because it has been linked to a number of outcomes that are problematic from both individual and societal perspectives, such as addiction and criminal behaviour (*Pratt & Cullen, 2000*; *Perry & Carroll, 2008*; *Moffitt et al., 2011*; *Sharma, Markon & Clark, 2014*). Individual differences in impulsivity are likely to reflect multiple influences, from genetics (*Kreek et al., 2005*), to early developmental factors (*Pettersson et al., 2014*), to aspects of the immediate social context (*Kidd, Palmeri & Aslin, 2013*). Here, we focus on two factors that have recently been shown to be associated with impulsivity: socioeconomic position and negative mood. A substantial number of studies have presented evidence that individuals of lower socioeconomic position tend to be more impulsive than those of higher socioeconomic position (*Lawrance, 1991*; *Green et al., 1996*; *Adams & White, 2009*). Though causality

Corresponding author
Daniel Nettle,
daniel.nettle@ncl.ac.uk

is difficult to establish definitively, the relationship has been argued to be at least partly causal (*Haushofer & Fehr, 2014*). That is, experiencing socioeconomic hardship may make people become more impulsive, rather than, for example, more impulsive people being downwardly economically mobile. Other studies have shown that negative mood can also make people more impulsive (*Lerner, Li & Weber, 2012*). Here, the causality is much easier to demonstrate, since the negative mood was induced experimentally through watching a film clip about a death (or a neutral control film clip) prior to the impulsivity task. The socioeconomic and mood effects on impulsivity may well be linked; part of the reason that people of lower socioeconomic position are characterised as more impulsive may be that their mood is more negative much of the time (*Haushofer & Fehr, 2014*). However, whether there is also a 'mood-independent' effect of socioeconomic position on impulsivity is not at present well understood.

Impulsivity, however, is not a unitary construct (*Reynolds et al., 2006*; *Sharma, Markon & Clark, 2014*; *Stahl et al., 2014*). Although a number of typologies of impulsivity have been proposed, a common distinction is between the unwillingness to wait for a deferred outcome ('waiting impulsivity' or impulsive choice) and the inability to stop oneself from making a response that has been cued or initiated by the context ('stopping impulsivity,' impulsive action, or behavioural disinhibition) (*Reynolds et al., 2006*; *Perry & Carroll, 2008*; *Brevers et al., 2012*). Although both types are often referred to as impulsivity in the literature, measures of waiting impulsivity do not tend to be substantially correlated with measures of behavioural disinhibition (*Reynolds et al., 2006*). Almost all of the evidence on socioeconomic position and current mood as predictors of impulsivity comes from waiting impulsivity tasks. For behavioural disinhibition, one study found no socioeconomic patterning in a large community sample of children and adolescents (*Crosbie et al., 2013*). Beyond this, there is little research looking for a socioeconomic gradient, and there have been no experimental studies manipulating mood to examine the consequences for behavioural disinhibition. In this study, we therefore administered a standard measure of behavioural disinhibition, the stop-signal task, to adults from a community sample, having first measured their socioeconomic background, and randomly assigned them to receive either a negative mood induction or a control procedure.

The stop-signal task measures behavioural inhibition by presenting participants with two concurrent tasks, the go task and the stop task. The go task is a visual identification task to be completed as rapidly as possible. The stop task involves presentation of an additional cue (the stop signal) that tells the participant not to complete the go task on that particular trial. The stop signal occurs on only a minority of trials. Thus, the participant forms a prepotent response pattern of completing the go task on every trial, but on a minority of trials has to inhibit that prepotent response. Whether she successfully inhibits the response depends on which process completes first: the prepotent impulse to go, or the inhibitory impulse produced in response to the stop signal (*Logan & Cowan, 1984*; *Logan, Cowan & Davis, 1984*). The temporal offset between the appearance of the go-signal and the appearance of the stop-signal is varied in order to identify, for each participant, the offset at which the stop impulse completes sooner than the go impulse more often than

not. This makes it possible to assign each participant a stop-signal reaction time (SSRT). This is in effect a measure of how long it takes a person to mount an inhibitory response to a prepotent action impulse. A longer SSRT equates to poorer behavioural inhibition (or equivalently, greater behavioural disinhibition). SSRT scores have been validated as measures of stopping impulsivity in studies of drug addiction (*Fillmore et al., 2002*). When analysing SSRTs, it is important to control for participants' average reaction time on the go trials (the go reaction time, or GRT), since some of the variance in SSRT may be explained by how slow the go response is rather than how fast the inhibitory response is. Both SSRT and GRT have also been shown to increase with age in adulthood (*Williams et al., 1999*; *Bedard et al., 2010*).

There are multiple ways of conceptualizing and assessing socioeconomic position. Recent studies have suggested that exposure to deprived neighbourhoods across childhood may be a key predictor of psychological outcomes (*Sampson, Sharkey & Raudenbush, 2008*; *Sastry & Pebley, 2010*; *Sharkey & Elwert, 2011*). Thus, we chose to focus on neighbourhood socioeconomic deprivation rather than individual socioeconomic status. We therefore decided to use residential postcodes to obtain a deprivation score for the neighbourhood in which the participant had grown up. The UK has high-quality publicly-available neighbourhood deprivation data resolved to a small spatial scale, based on the average of indices across multiple domains of deprivation.

For the mood induction, we used the Velten procedure, a widely-used technique where the participant reads a set of written statements (*Velten, 1968*). Velten created three such sets, matched for linguistic properties: one emotionally neutral, one aimed to produce a depression-like negative mood, and one aimed to produce an elated positive mood. The Velten statements have been widely used, and previous studies have shown that participants' self-rated mood is more depressed and more anxious following the negative than the control set (*Kenealy, 1986*; *Westermann, Stahl & Hesse, 1996*). Thus, the effect of the manipulation appears to be both a reduction in positive affect and an increase in negative affect (*Watson & Tellegen, 1985*). Although many studies have found significant effects of the Velten mood induction procedure on mood, it is important to include a manipulation check to verify this in each experiment, since a null finding in relation to the outcome of interest is uninterpretable unless the mood manipulation is known to have worked (*Kenealy, 1986*).

Our predictions were that participants from more deprived backgrounds, and participants assigned to the negative condition, would show relatively longer SSRTs, once appropriate control was made for age and GRT. We noted the possibility that there might also be interactions between socioeconomic background and mood condition, since some recent studies have suggested that people from different socioeconomic backgrounds react differently to cues of current adversity (*Griskevicius et al., 2013*).

**Peer**J

## METHODS

### Ethics and participants

The study was authorised by the Newcastle University Faculty of Medical Sciences Ethics Committee under approval number 00655/2013. Participants were an opportunity sample of individuals who had grown up in the UK, recruited by means of the Institute of Neuroscience participant pool, Newcastle University. This is a large database of email addresses of people who have shown an interest in taking part in neuroscience or medical research. It includes students, staff of the university, and other residents of the city, and is thus reasonably diverse in terms of ages and socioeconomic backgrounds. A compensation of £5 was offered in exchange for participation. The data includes two samples gathered in separate years by Thomas Carpenter and Tünde Paál, respectively. All procedures were identical in the two sub-samples. We have repeated all analyses in this paper with experimenter as an additional random effect, and none of the results is altered. A total of 58 people participated (65.5% female; age (in years) $M = 32.77$, $SD = 14.8$).

### *Procedure*

Participants were tested individually in a single session in a curtained cubicle within a computer laboratory. The experimenter withdrew from the cubicle during the tasks. On a desk in front of the participant was a desktop computer with speakers and keyboard. Standard computer keyboards have slow polling rates that render them unsuitable for tasks requiring highly accurate timing (*Plant, Hammond & Whitehouse, 2003*; *Verbruggen, Logan & Stevens, 2008*), and so we used a Razer 'Lycosa' games keyboard with a specified polling rate of 1,000 Hz. The sequence of steps was as follows: participants read the study information sheet and signed a consent form; completed a computerised demographic questionnaire and baseline mood measure (using the Qualtrics survey platform, www.qualtrics.com); completed the mood induction task; completed the stop-signal task; and finally completed another mood measure as a manipulation check. On completion, each participant was debriefed and received the compensation payment.

*Mood induction procedure.* After completion of the demographic questionnaire, participants were pseudorandomly assigned to either the negative or neutral conditions. This was done automatically by the computer and the experimenter did not know which the experimental condition the participant was completing. Participants in the negative condition were shown a sequence of 50 statements from the depression Velten set (*Velten, 1968*), whereas participants in the neutral condition saw 50 from the neutral set. Each statement was presented on the screen for six seconds, and the participants were instructed to try to memorise them. After the mood induction had finished, the participant alerted the experimenter, who started up the stop-signal task programme.

*Stop signal task.* We delivered the stop-signal task using STOP-IT software (*Verbruggen, Logan & Stevens, 2008*). In this implementation, the go stimulus is a square or circle displayed in the middle of the screen; two response keys on the keyboard were marked with a square and circle respectively, and the participant instructed to press the correct one as quickly as possible. The stop signal is an audible tone. STOP-IT was run in

full-screen mode, with system volume set to full and speaker volume at 2/3. The default task parameters were used: 1 practice block of 32 trials, followed by 3 experimental blocks of 64 trials each. Each trial begins with a 250 ms fixation cue, followed by the go stimulus, which is displayed until the participant responds, with a maximum limit of 1250 ms. The inter-trial interval is 2000 ms. Stop-trials constitute 25% of all trials; the difference in onset time of stop-cue relative to go-cue on these trials is automatically titrated dependent on performance to provide an estimate of the delay at which that participant has a 50% chance of inhibiting successfully. The participant's SSRT is calculated from the value of this delay.

After the experimenter had explained the instructions, participants pressed a key when they were ready to start the task. The experimenter observed the participant during the practice block to see whether they were responding correctly. The participant then completed the 3 experimental blocks, which the experimenter did not observe. Following the practice block and each one of the trial blocks, a summary screen showing the participant's response suppression rate, trials missed and errors made was displayed during a 10-s delay until the participants could press a key to begin the next task.

### Measures

*Baseline and final mood.* Baseline and final mood were rated with a single item, on a scale from 1 (the most negative mood possible) to 100 (the most positive mood possible).

*Neighbourhood deprivation score.* We calculated a neighbourhood deprivation score using the postcode from age 16 provided by the participant. The age of 16 was chosen somewhat arbitrarily but seemed likely to be one the participant would remember as well as providing a reasonable summary measure of the kind of socioeconomic environment in which they had grown up. The UK Neighbourhood Statistics database (https://neighbourhood.statistics.gov.uk/) allows each UK postcode to be assigned a rank in terms of the deprivation of the census tract (lower super output area) in which it is found, from 1 (most deprived tract in the UK) to 32,482 (least deprived in the UK). To make these ranks more intuitively interpretable, we rescaled them using the formula:

Deprivation score $= 1 - $ (deprivation rank/32,482).

Thus, a postcode in the median UK neighbourhood would have a deprivation score of 0.5, the most deprived a score of 1, and the least deprived a score of 0.

*Age.* Age was given in years in the initial demographic questionnaire.

*SSRT and GRT.* SSRT and GRT were calculated using the ANALYZE-IT programme supplied with STOP-IT (*Verbruggen, Logan & Stevens, 2008*). All participants provided data that met the assumptions required by STOP-IT to generate a valid SSRT and GRT.

### Data analysis

Once the measures had been calculated, data were analysed in R (*R Core Development Team, 2013*), using general linear models and *t*-tests as appropriate. The structure of each general linear model we ran is described in the Results section. SSRTs were mildly right-skewed. All models reported below were also run using log-transformed SSRT, which corrects the right skew. Since results were essentially identical, the non-transformed results

**Table 1 Descriptive statistics by condition for age, baseline mood and deprivation score.**

| Participants | Negative 30 (18 female) | Neutral 28 (20 female) | Condition difference |
|---|---|---|---|
| Age (years) | $M = 34.8, SD = 15.9$ | $M = 30.5, SD = 13.3$ | $t_{55} = 1.09, p = 0.28$ |
| Baseline mood (1–100) | $M = 80.4, SD = 17.9$ | $M = 71.7, SD = 22.6$ | $t_{56} = 1.62, p = 0.11$ |
| Deprivation score (0–1) | $M = 0.45, SD = 0.29$ | $M = 0.45, SD = 0.25$ | $t_{55} = 0.09, p = 0.93$ |

have been reported. Residuals were checked in all cases, and there were no major violations of parametric model assumptions.

## RESULTS

Raw data are available as Supplemental Information 1.

### Descriptive statistics

Table 1 shows the descriptive statistics by condition for variables other than performance in the stop-signal task. There were no significant differences by condition in age, deprivation, or baseline mood, as there should not have been, since condition was randomly assigned. Average deprivation scores were close to the UK median neighbourhood (0.45), but with a good range (0.11–0.98). Deprivation score was not significantly associated with age ($r_{55} = 0.18, p = 0.18$), or with baseline mood ($r_{56} = 0.07, p = 0.62$).

### Mood manipulation and final mood

The mood induction produced a modest decrease in mood between baseline and final rating in the negative condition ($M = -3.23; SD = 5.27$), and a modest improvement in mood in the neutral condition ($M = 2.78; SD = 13.31$). The condition difference in mood change was statistically significant ($t_{56} = -2.29, p = 0.03$). However, the final mood ratings were not significantly different between the negative and neutral conditions (Negative: $M = 77.1, SD = 18.1$; Neutral: $M = 74.5, SD = 17.3, t_{56} = 0.56, p = 0.58$). This is due to variation in initial mood diluting the modest effect of the mood manipulation. For this reason, in all subsequent analyses, we include both experimental condition and baseline mood as independent variables. We also repeated the analyses presented below replacing experimental condition and baseline mood with final mood in the models. No effects of final mood were significant, and no conclusions were changed. Hence, these models are not described further.

### Go-reaction time (GRT)

We fitted a general linear model to the GRTs, with age, initial mood, condition, deprivation score and the condition by deprivation score interaction as predictors. The model summary is shown in the upper part of Table 2. Age predicted GRT, with GRTs becoming slower with increasing age, but there were no significant effects of initial mood, condition or deprivation score on GRT.

**Table 2  Summary of general linear models predicting go reaction time in the stop-signal task (GRT; upper rows) and stop-signal reaction time in the stop-signal task (SSRT; lower rows).**

| Variable | Parameter estimate | Standard error | $t$ | $p$ |
|---|---|---|---|---|
| **Outcome variable: GRT** | | | | |
| Age | 4.85 | 1.35 | 3.59 | 0.01 |
| Deprivation score | 185.20 | 225.82 | 0.82 | 0.42 |
| Baseline mood | 0.46 | 0.98 | 0.47 | 0.64 |
| Condition | 8.39 | 76.27 | 0.11 | 0.91 |
| Deprivation score ∗ condition | −117.44 | 150.16 | −0.78 | 0.44 |
| **Outcome variable: SSRT** | | | | |
| GRT | −0.08 | 0.04 | −2.15 | 0.04 |
| Age | 1.08 | 0.41 | 2.63 | 0.01 |
| Deprivation score | 146.65 | 61.83 | 2.37 | 0.02 |
| Baseline mood | −0.40 | 0.27 | −1.50 | 0.14 |
| Condition | 18.40 | 20.75 | 0.89 | 0.38 |
| Deprivation score ∗ condition | −65.26 | 41.09 | −1.59 | 0.12 |

## Stop-signal reaction time (SSRT)

The general linear model for SSRT included as predictors GRT, age, initial mood, condition, deprivation score, and the condition by deprivation score interaction. The model summary is shown in the lower part of Table 2. There was a significant effect of GRT, with participants who were faster on the go-trials having a longer SSRT. There was also a predicted effect of age, with older participants having longer SSRTs. There was a significant effect of deprivation score, with participants from more deprived neighbourhoods having longer SSRTs (Fig. 1). The effects of initial mood, condition, and the condition by deprivation interaction were not significant.

## DISCUSSION

In a community sample of adults, we found evidence that behavioural disinhibition, as measured using the stop-signal task, was related to socioeconomic background. Participants who had lived in more deprived neighbourhoods at age 16 had longer SSRTs, and hence showed greater behavioural disinhibition, than participants who had lived in less deprived neighbourhoods. We also measured mood, and manipulated it using a standard mood-induction procedure, but we found no evidence that current mood—either the naturally-occurring variation in baseline mood, or our experimentally-produced mood manipulation—affected stop-signal task performance in any way. Thus, our predictions concerning the relationship between socioeconomic deprivation and behavioural disinhibition were supported, whilst our predictions concerning the role of current mood were not. Our results concurred with those of previous investigations in finding that SSRTs, as well as GRTs, increased substantially with age (*Williams et al., 1999*; *Bedard et al., 2010*). We also found that SSRT was negatively related to GRT; people who were faster to act in the go-trials were, other things being equal, slightly more disinhibited.

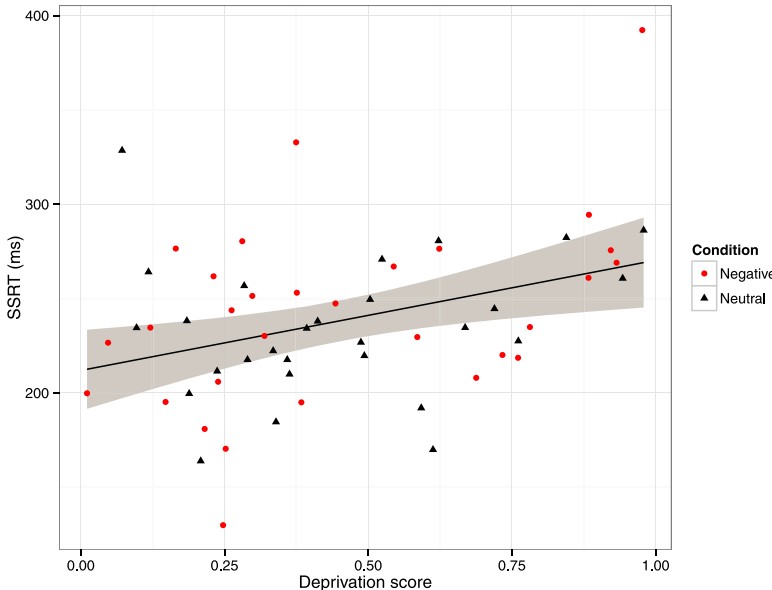

**Figure 1** Scatterplot of stop-signal reaction time (SSRT) against deprivation score, with participants labelled by condition in the mood manipulation.

However, the deprivation-disinhibition relationship was not explained by differences in GRT; it persisted even once variation in GRT was controlled for.

A number of previous studies have demonstrated socioeconomic gradients in measures of 'waiting' impulsivity, the relatively steep devaluing of future rewards compared to immediate ones (*Lawrance, 1991*; *Green et al., 1996*; *Adams & White, 2009*). To our knowledge, this is the first evidence that there may also be a socioeconomic gradient in 'stopping impulsivity' or behavioural disinhibition. Demonstrating the existence of such a gradient is potentially important, since behavioural disinhibition predicts problematic real-world outcomes above and beyond 'waiting' impulsivity alone (*Brevers et al., 2012*; *Sharma, Markon & Clark, 2014*). Our results differ from those of the most comparable previous study (*Crosbie et al., 2013*), who found no associations between socioeconomic deprivation and stop-signal performance in a large Canadian community sample. Crosbie et al. used a similar approach to assessing deprivation as ours, using residential postcodes. They report that their sample was skewed towards more economically privileged neighbourhoods (p. 501), though direct comparison with the composition of our sample is not straightforward due to the differences in measures and in countries. Crosbie et al. also studied children (mean age 11), whereas we studied adults.

Why socioeconomic gradients in impulsivity should exist is not well understood. One recent proposal is that relationships between socioeconomic variables and impulsivity might be mediated by differences in mood (*Haushofer & Fehr, 2014*). However, this cannot be the case here, since there was no socioeconomic gradient in baseline mood, and we found no evidence for any effect of mood on behavioural disinhibition. This is in contrast to a previous experimental study that found effects of mood manipulation on waiting impulsivity (*Lerner, Li & Weber, 2012*). Thus, our data suggest an embedded

effect of socioeconomic deprivation on disinhibition that is independent of current mood. Socioeconomic deprivation is associated with many different aspects of experience, and exactly what it is about deprivation that tends to produce greater disinhibition as well as a relatively stronger preference for immediate over delayed outcomes remains to be explored.

Our study had a number of limitations. Our sample was not constructed in such a way as to guarantee socioeconomic representativeness. However, we were fortunate in that the mean deprivation score of our sample was roughly in the middle of the UK range, and both extremes were represented in the data. We measured deprivation through postcode at age 16. We chose this as we anticipated recruiting mainly young adults, and we felt this would be the single most convenient and effective measure for this age group. In fact, we recruited more, older people than anticipated, and childhood postcode at 16 is a less ideal measure for these participants than for younger ones. For one thing, their age 16 is longer ago, and the neighbourhoods may have changed in the intervening years. We made no attempt to distinguish statistically between deprivation experienced whilst growing up and adult deprivation. It could be that the most important predictor of behavioural disinhibition is *current* experience of deprivation (*Nettle et al., 2014*), and the relationship we found with deprivation at age 16 might be because people with more deprived childhood and adolescent backgrounds also go on to experience more deprivation in adulthood. To discriminate whether current or developmental experience of deprivation is important, the two would need to be separately measured to establish which has the greater predictive power. Our measure also did not distinguish individual-level socioeconomic characteristics such as parental income and education from neighbourhood-level deprivation. The two are likely to be highly correlated, but our study does not licence inferences about which of these, if either, is responsible for the observed association.

We did not measure current education level or any measure of general cognitive ability. This means we cannot exclude the possibility that general cognitive ability or education level mediates the relationship between socioeconomic background and behavioural disinhibition. There was no relationship between deprivation score and GRT, suggesting that the association was specifically with disinhibition, rather than with task performance more generally, but without a greater range of control tasks we are unable to make strong inferences on this point. More generally, caution is required about inferring causality from an association; though it seems reasonable that something about childhood deprivation might cause the development of greater disinhibition, other explanations for the correlation cannot be excluded.

Our mood manipulation, though it employed a standard technique that has been used in other recent studies (*Smallwood & O'Connor, 2011*; *Scherrer, Dobson & Quigley, 2014*), produced only very modest effects on participants' final mood. It is possible that a stronger manipulation of mood might have affected behavioural disinhibition. However, there was considerable variation in baseline mood, and the combination of baseline mood and the manipulation still did not explain any variation in behavioural disinhibition. This suggests

that across a substantial range of mood, mood effects on behavioural disinhibition, if they exist, must be very small.

Despite the limitations noted above, it is striking that in a relatively small sample, and with a relatively crude measure of socioeconomic background, we found evidence of an association between the experience of deprivation and behavioural disinhibition. Childhood socioeconomic deprivation is an epidemiological predictor of a number of important outcomes such as subsequent crime (*Levine, 2011*), and transition to habitual use of drugs and tobacco (*Legleye et al., 2011*). Behavioural disinhibition plausibly plays a role in these outcomes. Thus, behavioural disinhibition could be an important psychological mechanism linking childhood socioeconomic conditions to subsequent life outcomes.

### Funding

This research was supported by the Graduate School of the Faculty of Medical Sciences, Newcastle University. Tünde Paál was supported by an Eötvös Hungarian State Scholarship. The funders had no role in study design, data collection and analysis, decision to publish, or preparation of the manuscript.

### Grant Disclosures

The following grant information was disclosed by the authors:
Graduate School of the Faculty of Medical Sciences, Newcastle University.
Eötvös Hungarian State Scholarship.

### Competing Interests

The authors declare there are no competing interests.

### Author Contributions

- Tünde Paál conceived and designed the experiments, performed the experiments, analyzed the data, wrote the paper, prepared figures and/or tables, reviewed drafts of the paper.
- Thomas Carpenter conceived and designed the experiments, performed the experiments, analyzed the data, wrote the paper, reviewed drafts of the paper.
- Daniel Nettle conceived and designed the experiments, analyzed the data, wrote the paper, prepared figures and/or tables, reviewed drafts of the paper.

### Human Ethics

The following information was supplied relating to ethical approvals (i.e., approving body and any reference numbers):

The study was authorised by the Newcastle University Faculty of Medical Sciences Ethics Committee under approval number 00655/2013.

## Supplemental Information

Supplemental information for this article can be found online at http://dx.doi.org/10.7717/peerj.964#supplemental-information.

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
