# Peer review of "Childhood socioeconomic deprivation, but not current mood, is associated with behavioural disinhibition in adults"

_PeerJ, doi:10.7717/peerj.964_

## Round 0.1 · original submission · Minor Revisions

Dear Authors,

Overall I think the article represents a positive contribution to the literature and think there should be moderate revisions based on the independent reviewers comments. I agree with all the comments and they should each be addressed. I would like to particularly highlight a couple comments.

I believe a demographic table would be very helpful. The reason is I think, there should be more emphasis on additional variables associated with disinhibition when possible (understand you do not have all of them). This is partially because you are comparing state mood induction with static environmental variables. Because of the lack of findings regarding mood, more emphasis might need to be placed on these environmental/personal variables. Ideally I would add any variables you have into the model to further understand the relationships. With this, I believe the reviewers comments regarding that the discussion should be must more conservative regarding the results and implications should be taken into account.

Overall, I believe this is an important contribution to the literature and should be published if the reviewer comments are addressed appropriately. Thanks

Reviewer 1 ·

Basic reporting

The introduction needs to be highly improved.
Methodology should be improved by adding a great part of the introduction and a clear "measure or material" bullet would be helpful for the reader.

Experimental design

no comments

Validity of the findings

In the first paragraph of the discussion, authors should be more conservative on the results found in the current study. Some limitations, such as the very small sample size, do not allow to have robust conclusion, but only speculations of possible relationships.

Additional comments

The manuscript should be improved. At the moment it is quite disorganised, with important parts written in the wrong sections.

Annotated reviews are not available for download in order to protect the identity of reviewers who chose to remain anonymous.

·

Basic reporting

A demographics table might be helpful, containing n's for each condition (neutral and negative mood), age, and deprivation score (mean, SD for each) for both men and women separately in each condition. Other than that, I think that overall, the study reads clearly and has a logical flow throughout the manuscript.

Experimental design

No Comments - all limitations are mentioned in the discussion.

Validity of the findings

In Figure 1, it appears that there is an outlier near the highest deprivation category (1.0) and was wondering if the correlation was dependent upon this individual. Additionally, outside of the small group near the highest deprivation categories, does the correlation still hold for those between 0.0 and 0.75?

When reading the study, the questions that I had were largely related to the mood component having a strong enough effect to be measured in the context of the inhibitory task. Additionally, I was concerned about the variation in age impacting socioeconomic status at 16 (the variation in the time since living in that socioeconomic region) and the lack of measures of current education level, IQ, or current socioeconomic status. The authors do address many of these concerns in the discussion. I do think more emphasis/explanation might be helpful for each of these areas. For example, the authors mention "general cognitive ability", but this measure is not necessarily related to the GRT, and education level and IQ have been implicated in mediating task performance in various inhibitory or interference tasks. Again, education, IQ, and current socioeconomic status are generally linked to childhood environment and I think a discussion covering how various factors can influence socioeconomic status (there is some mention of this, but I think it can be elaborated upon). Finally, I think that the authors might consider discussing possibilities for why their results differ from those of Crosbie et al 2013 mentioned in the introduction (outside of mood).

Additional comments

I think that the study provides some interesting findings regarding behavioral disinhibition that warrant further study.

---

## Round 0.2 · accepted · Accept

The current changes to the document are satisfactory and I think the paper adds to the literature. Thank you for submitting. Please make the following change before finalizing with PeerJ staff. Reviewers including myself were looking for more demographics (anything) to make a true demographics table with multiple variables. At no point until the end is that you only have 3 variables discussed. Please include a statement indicating that age, gender and deprivation score were the only demographic variables collected. Please state this upfront in the methods and change the name demographic questionnaire to baseline questionnaire. Having the term demographic questionnaire misleads the reader to expect more but is not a weakness. I look forward to seeing the final and incorporating it into my work. Thanks for your efforts.